

# Atomistic and coarse grained simulations reveal increased ice nucleation activity on silver iodide surfaces in slit and wedge geometries

Golnaz Roudsari[1], Olli H. Pakarinen[1], Bernhard Reischl[1], and Hanna Vehkamäki[1]

[1]Institute for Atmospheric and Earth System Research / Physics, University of Helsinki, PO Box 64, FI-00014, Finland

**Correspondence:** Golnaz Roudsari (golnaz.roudsari@helsinki.fi)

**Abstract.** Ice clouds can form at low and moderate supercooling through heterogeneous ice nucleation on atmospheric particles. Typically, the nucleation requires active sites with special chemical and physical properties, including surface topology and roughness. This paper investigates microscopic mechanisms of how combinations of confinement by the surface topology and lattice match induced by the surface properties can lead to enhanced ice nucleation. We perform molecular dynamics simulations using both atomistic and coarse-grained water models, at very low supercooling, to extensively study heterogeneous ice nucleation in slit-like and concave wedge structures of silver-terminated silver iodide (0001) surfaces. We find that ice nucleation is greatly enhanced by slit-like structures when the gap width is a near-integer multiple of the thickness of an ice bilayer. For wedge systems we also do not find a simple linear dependence between ice nucleation activity and the opening angle. Instead we observe strong enhancement in concave wedge systems with angles that match the orientations of ice lattice planes, highlighting the importance of structural matching for ice nucleation in confined geometries. While in the slit systems ice cannot grow out of the slit, some wedge systems show that ice readily grows out of the wedge. In addition, some wedge systems stabilize ice structures when heating the system above the thermodynamics melting point. In the context of atmospheric ice nucleating particles, our results strongly support the experimental evidence for the importance of surface features such as cracks or pits functioning as active sites for ice nucleation at low supercooling.

## 1 Introduction

Heterogeneous ice nucleation is ubiquitous and important in the atmospheric processes at temperatures above 235 K (Tabazadeh et al., 2002; Djikaev et al., 2002). In particular, the freezing of cloud droplets on an ice nucleation particle (INP) affects both the microphysical cloud properties as well as cloud albedo, which contributes to the radiative balance of the planet (Pruppacher and Klett, 2010; Kanji et al., 2017; Hawker et al., 2021). The formation of ice crystals on foreign surfaces is also relevant in biology (Christner et al., 2008; Ling et al., 2018), and materials science (Guoying et al., 2019). Whether or not a particle is ice nucleation active depends on its chemical and physical properties, including surface topology and roughness (Murray et al., 2012; Campbell et al., 2015; Zhang et al., 2020; Koop, 2017; David et al., 2019).

Classical nucleation theory (CNT) suggests that ice can nucleate from a smaller critical nucleus in concave cavities (Turnbull, 1950). Based on CNT, it is argued that irregular surfaces can promote heterogeneous ice nucleation (Marcolli, 2014; Kiselev





et al., 2017; Christenson, 2013; Zielke et al., 2016). A number of experimental and computational studies have explored the impact of nanometer-sized cavities, referred to as *confinement*, on ice nucleation (Limmer and Chandler, 2012; Moore et al., 2010; Holden et al., 2019, 2021; Kastelowitz and Molinero, 2018). On the other hand, it has been discovered that the nucleation of ice can be enhanced by some specific geometrical structures at the surface of the material, also referred to as *lattice match* (Zielke et al., 2016; Fraux and Doye, 2014). In recent years, effects of combinations of confinement and lattice match (i.e.,

irregularities on crystalline surfaces with ice-like structures) on ice nucleation has received a great deal of attention (Kiselev et al., 2017; Holden et al., 2019; Campbell et al., 2017, 2015; Hiranuma et al., 2014; Holden et al., 2021; Li et al., 2018; Bi et al., 2017).

   Experimental techniques have been widely used to identify the *active sites* for heterogeneous ice nucleation on minerals, such as feldspars, which are considered to be the single most important group of mineral dusts for atmospheric ice nucleation. For

example, Kiselev et al. (2017) showed that the ice nucleating sites on the K-feldspar surface consist mainly of steps and patches with (100) plane structures. Another study found that micron-sized pits on the surface of K-feldspar significantly accelerate ice nucleation (Holden et al., 2019). In addition, it was shown that acute wedges of mica can enhance ice nucleation (Campbell et al., 2017). Other experimental works include studies of the effects of irregularities on the surfaces of hematite, silicon and glass on heterogeneous ice nucleation (Campbell et al., 2015; Hiranuma et al., 2014).

However, experiments are limited in spatial and temporal resolution and typically cannot provide insight on the atomistic details of the ice nucleation mechanism (Sosso et al., 2016). Classical nucleation theory is based on bulk thermodynamic properties and, in its heterogeneous variant, characterizes the effect of the INP by the macroscopic contact angle between a hemispherical ice crystal and a flat surface ignoring the atomistic details of the ice nucleus and the INP surface. Computer simulations are therefore widely used to provide insight on the microscopic mechanism of ice nucleation (Sosso et al., 2016).

For instance, Page and Sear (2006) studied the effect of slit pores with different widths on heterogeneous nucleation based on the Ising model. Later, they investigated the role of specific geometrical structures (wedges with varying angles) using a Lennard-Jones model (Page and Sear, 2009). Furthermore, the mW model of water (Molinero and Moore, 2009) has been used to study heterogeneous ice nucleation at nanogrooves with different widths on the surface of platinum (Zhang et al., 2014) and graphene (Li et al., 2018). Bi et al. (2017) studied ice nucleation in wedge structures of graphene using the coarse-grained mW

model at temperatures between 230 K and 240 K, and observed that nucleation rates are increased for opening angles of the wedge system equal to the dihedral angles of cubic ice planes.

   In this work, we employ molecular dynamics (MD) simulations using both TIP4P/Ice (Abascal et al., 2005) and mW water models to extensively study heterogeneous ice nucleation in slit-like and concave wedge structures of silver iodide in the wurtzite structure ($\beta$-AgI), exposing the Ag-terminated (0001) surface. It has been shown both experimentally and computa-

tionally that silver iodide is an effective material in promoting ice nucleation (Marcolli et al., 2016; Fraux and Doye, 2014; Shevkunov, 2016; Zielke et al., 2014; Prerna et al., 2019; Roudsari et al., 2020), and it has been used as a rain seeding agent for many decades (Vonnegut, 1947). Several studies have reported that the lattice match between the (0001) plane of AgI and a basal plane (0001) of hexagonal ice (ice I$_h$) or the (111) plane of cubic ice (ice I$_c$) leads to ice nucleation and growth occurring on time scales readily accessible to unbiased MD simulations at temperatures below about 260 K, making it a convenient





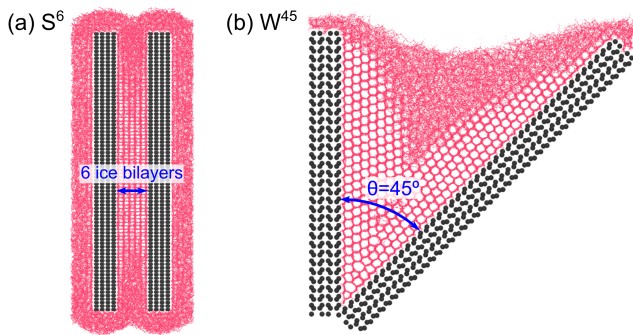

**Figure 1.** Simulation models of (a) slit system with a gap width accommodating 6 ice bilayers ($S^6$) and (b) wedge system with angle $\theta = 45°$ ($W^{45}$). AgI surface atoms are shown as black spheres and the hydrogen bond network between water molecules is shown as red sticks.

model system for *in silico* studies. (Zielke et al., 2014; Fraux and Doye, 2014). Unlike e.g. the previously studied graphene wedges (Bi et al., 2017), the AgI(0001) surface acts as a strong template for water molecules, imposing additional structure on ice growing in the spatial confinement of slit or wedge geometries. Studying a highly ice nucleation active material allows us to determine whether ice nucleation can be enhanced by confinement also at very low supercooling, i.e. above 263 K.

The remainder of this paper is organised as follows: Section 2 describes the simulation methodology, and slit and wedge system models. The simulations of ice nucleation in slit and wedge systems are extensively analysed in Sect. 3 and 4, respectively. Finally, in Sect. 5 we summarize our results and conclude this paper. Throughout the paper, the notation $S^d$ is used to denote a slit system with a gap width accommodating a number of $d$ ice bilayers, and $W^\theta$ is used to denote a wedge system with angle of $\theta$ degrees, as shown in Fig. 1

## 2 Methods

### 2.1 Force Fields

The mW model (Molinero and Moore, 2009) uses a coarse-grained representation where the water molecule is represented by a single particle, interacting through a short-range Stillinger-Weber potential only. Thus, simulations using this model are computationally cheaper compared to the all-atom TIP4P/Ice model (Abascal et al., 2005) which consists of four interaction sites with a rigid geometry and additionally requires the calculation of long-range Coulomb interactions between partial charges. We therefore employed the mW model to systematically screen a large number of systems for their ice nucleation activity, and then validated our observations on a relevant subset of these systems using the more expensive TIP4P/Ice model, which more accurately reproduces the properties of bulk water as well as the phase diagram (Abascal et al., 2005). The melting point is $270 \pm 3$ K for the TIP4P/Ice model (Fernandez et al., 2006), and $273.0 \pm 0.5$ K for the mW model (Hudait et al., 2016).

The slabs of silver iodide from which the slit and wedge geometries were constructed were modeled with ions fixed at the bulk lattice positions of a wurtzite $\beta$-AgI crystal with lattice constants $a = b = 0.458$ nm and $c = 0.75$ nm. The potential by




Hale and Kiefer (1980) was used to describe the interaction between $Ag^+$ and $I^-$ ions and TIP4P/Ice water molecules. Since TIP4P/Ice has a rigid point charge geometry, the polarization term in the Hale and Kiefer Force field was also ignored (Zielke et al., 2015). The ions were represented by point charges and an additional Lennard-Jones potential between the ions and the oxygen atom in water (OW). The Lennard-Jones parameters are $\epsilon_{Ag-OW} = 2.289$ kJ/mol, $\sigma_{Ag-OW} = 0.317$ nm for Ag-

OW interactions, and $\epsilon_{I-OW} = 2.602$ kJ/mol, $\sigma_{I-OW} = 0.334$ nm for I-OW interactions. The partial charge value is $\pm 0.6$e. However, using partial charges ranging from 0 to $\pm 0.8$e was shown to have no considerable impact on the ice nucleating characteristics of a flat silver iodide surface (Zielke et al., 2015, 2016). The cutoffs for Lennard-Jones and real-space Coulomb interactions were 0.85 nm.

For the mW water model simulations, interactions between the coarse-grained water molecule and $Ag^+$ and $I^-$ ions are

described by a Lennard-Jones potential with parameters fitted to reproduce both the exact corrugations of the water molecules in the first hydration layer over a flat AgI(0001) surface at 273 K, and the interfacial energy of a single hydration layer on the same surface, calculated using the atomistic TIP4P/Ice model, leading to parameters $\epsilon_{Ag-mW} = 2.81805$ kJ/mol, $\sigma_{Ag-mW} = 0.29469$ nm and $\epsilon_{I-mW} = 3.2041$ kJ/mol, $\sigma_{I-mW} = 0.32103$ nm, with a cutoff set to 0.803 nm.

## 2.2 Molecular Dynamics simulations

Molecular dynamics (MD) simulations with the TIP4P/Ice water model were performed using the GROMACS code (van der Spoel et al., 2005; Berendsen et al., 1995). The equations of motion are integrated using the velocity-Verlet algorithm with a time step of 2 fs. The long-range electrostatic interactions were calculated using the particle mesh Ewald (PME) method (Essmann et al., 1995). The bond lengths and angles of the TIP4P/Ice molecule were constrained using the LINCS algorithm (Hess et al., 1997). The MD simulations with the mW water model were performed using the LAMMPS code (Plimpton, 1995), using

the velocity-Verlet algorithm with a time step of 5 fs. All MD simulations were performed using periodic boundary conditions in three dimensions, with a vacuum gap between periodic images of the systems in $z$ direction. All the simulations were carried out in the NVT ensemble, controlling the temperature using a Nosé-Hoover thermostat (Nosé, 1984; Hoover, 1985) with a time constant of 0.4 ps and 1.0 ps for all-atom, and coarse-grained simulations, respectively.

## 2.3 Construction of slit and wedge systems

We studied the growth of ice in the slit geometry using a pair of mirrored Ag-terminated AgI(0001) slabs measuring $7.94 \times 18.09$ nm$^2$ in the $yz$-plane and thickness of 1.50 nm along $x$ coordinate, positioned at different distances from each other along the $x$ direction. The size of all simulation boxes in the $y$ direction is 7.4 nm, making the AgI slabs periodic in this direction. The simulation box sizes in the $x$ direction vary between 8 and 12 nm depending on the gap widths in the slit systems. In the $z$ direction, the box sizes vary between 30 and 35 nm, depending on the number of water molecules in the system. By breaking

the periodicity of the slabs along $z$ coordinate, we ensure that water molecules can quickly equilibrate to the correct density during ice formation. As ice forms bilayers of a well-defined thickness parallel to the flat AgI(0001) surface, as shown in Fig. 2, we considered gap widths ranging from 1.479 nm (4 ice bilayers) to 4.295 nm (12 ice bilayers), in steps of 0.0176 nm (1/20 of the width of an ice bilayer).





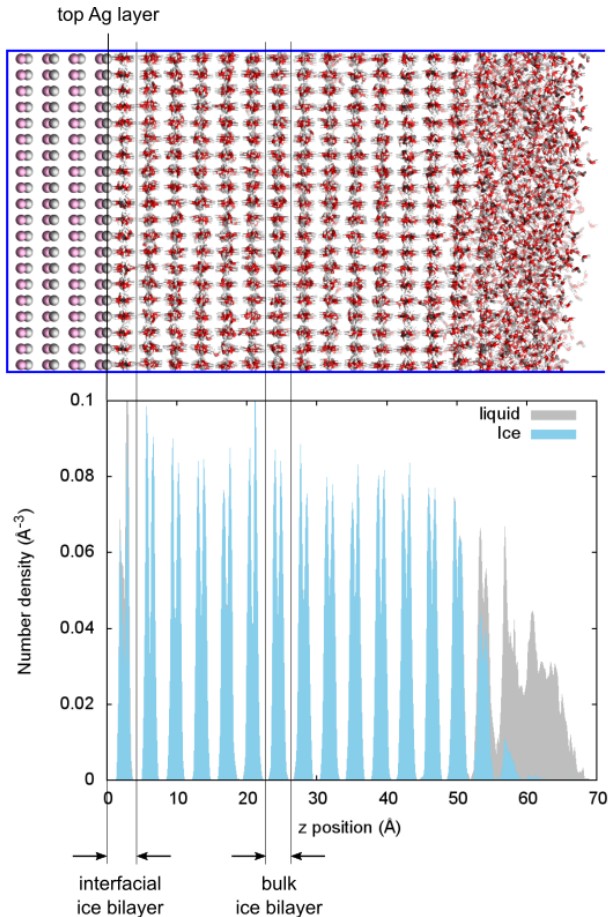

**Figure 2.** Top: snapshot of a simulation of ice growing in a film of water on a flat Ag-terminated AgI(0001) surface. Ag and I are shown in silver and pink, respectively, and water molecules are shown as red and white sticks. Bottom: number density profiles of water molecules along the z axis, perpendicular to the AgI surface. Ice and liquid water are shown in blue and gray, respectively, and the widths of an ice bilayer at the interface and in the bulk are indicated by black arrows.

We also investigated the growth of ice within wedges with twelve different angles $\theta$. To create the wedge systems, $W^\theta$, with wedge angles $\theta = 30, 45, 60, 70, 73, 110$ and $120$ degrees, two Ag-terminated (0001) surface slabs of $\beta$-AgI are connected along a line of Ag ions, along the y direction of the simulation box. To create the $W^{32}$ and $W^{62}$ wedge systems, a wedge-shaped cavity was cut out of a single $\beta$-AgI crystal, revealing one Ag-terminated (0001) surface and another regular Ag-terminated surface of higher index. The sizes of the simulation boxes along $x$ and $z$ coordinates varied between 9.45 and 23.95 nm, and 20 and 25 nm, respectively, according to the wedge angle, while the $y$ direction was 7.33 nm for each system built from two individual slabs, and 10.31 nm for the systems cut from a single crystal. All wedge systems are periodic along the $y$ direction. To generate the liquid water within the wedge configuration, the systems were first fully solvated in bulk liquid water, then water





molecules outside the wedges were removed. The number of water molecules in each system varies significantly depending on the wedge angle.

In the wedge systems, when using the TIP4P/ice model, the dipole moment from the two AgI(0001) surfaces is not fully

cancelled, unlike for the parallel, mirrored surfaces in the slit geometry (Roudsari et al., 2020). In the two systems cut out of a single crystal of $\beta$-AgI, the two sides of the wedge are not equal, enhancing the effect. However, near the bottom of the wedges where ice nucleation typically occurs, the effect of the dipole appears to be minor, based on the similarity of the results obtained with the TIP4P/Ice and mW model, the latter of which does not contain partial charges.

All systems were equilibrated at 273 K for several nanoseconds after solvation, before starting production runs at the desired

temperature. Fig. 1 shows two examples of the simulated slit and wedge systems.

## 3 Ice nucleation in AgI slit systems

We investigated the effects of various gap widths on ice nucleation and growth in slits between mirrored Ag-terminated AgI(0001) slabs at temperatures of 263 K, 265 K and 267 K. Simulation details are provided in Sec. 2.3. For each gap width, an independent MD simulation was carried out using the mW model, corresponding to a total of 161 simulations at each tem-

135 perature. For these systems, we define the nucleation rate as the inverse of the induction time, taken as the simulation time at which a continuous 'ice bridge' between the two slabs first appears. Once this happens, ice continues to grow rapidly. Using the mW model, we find that the nucleation rate exhibits periodic oscillations, with the maxima decreasing with increasing slit width. In particular, we observe that when the gap width is an integer multiple of the width of an ice bilayer (see Fig. 2), ice nucleation is significantly enhanced. This is enabled by a good *structural match*, i.e. the ice growth starting from the two sides

can combine together to form a continuous ice structure. In contrast, when the gap width of a slit system does not accommodate an integer number of ice bilayers, the nucleation of ice is strongly suppressed, up to gap widths of 12 ice bilayers.

To verify the above-mentioned observations, we conducted additional simulations using the TIP4P/Ice water model for 12 slit systems with gap widths between 4 and 12 times the thickness of an ice layer (each ice layer is ∼0.38 nm wide). Fig. 3 summarizes the nucleation rates observed using both mW and TIP4P/Ice water models as a function of the distance between

145 the two AgI slabs.

Fig. 4 shows the last frame of each simulation of slit systems with gap widths between 4 and 6.5 ice layers using the TIP4P/ice model, where the periodic enhancement/suppression effect is clearly visible. In the slit systems with wider gaps (e.g., more than 7 layers) the impact of confinement becomes insignificant. This can be explained by the fact that at these gap widths, the ice nucleation tends to start at each of the surfaces independently. For comparison, the nucleation rate at 263 K on

a single flat AgI(0001) surface using the TIP4P/Ice water model is $3.34 \times 10^{23}$ m$^{-2}$s$^{-1}$, as obtained from the value in Tab. 1 and shown in Fig. 3a, after dividing by the system's $x$ dimension. At 265 K and 267 K no nucleation is observed on a single flat AgI(0001) surface.

In previous work, Zhang et al. (2014) investigated the nucleation of ice on platinum surfaces with grooves of different widths using the mW model. They observed that ice nucleation is enhanced when the width of the grooves was near the ice lattice



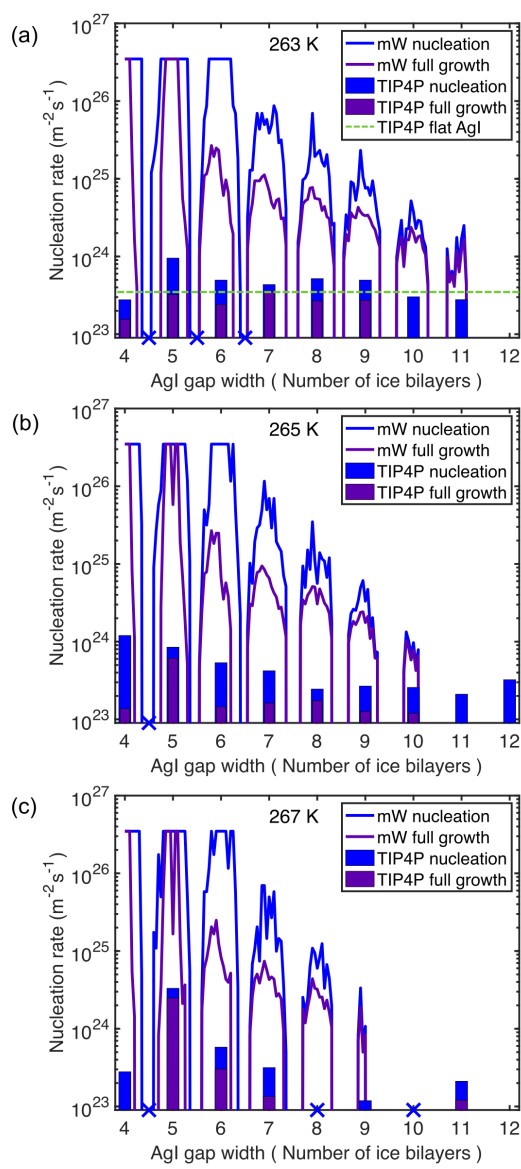

**Figure 3.** Nucleation rates in AgI slit simulations and inverse of time at which full ice growth in the AgI slit is observed, at (a) 263 K, (b) 265 K, and (c) 267 K, with mW model (continuous line) and TIP4P/Ice model (bar plot). Blue crosses indicate TIP4P/Ice simulations in which nucleation was not observed. The nucleation rate at 263 K on the flat AgI(0001) surface using the TIP4P/Ice model is indicated by the dashed green line in panel (a).

constant (0.743 nm). In a more recent work, Li et al. (2018) studied the effect of graphene nanogrooves on ice nucleation using the mW model. They also showed that the nucleation rate is affected by the width of the nanogrooves. Their simulations included 11 systems of different widths between 0.492 nm and 2.952 nm. They suggested that ice nucleation is substantially





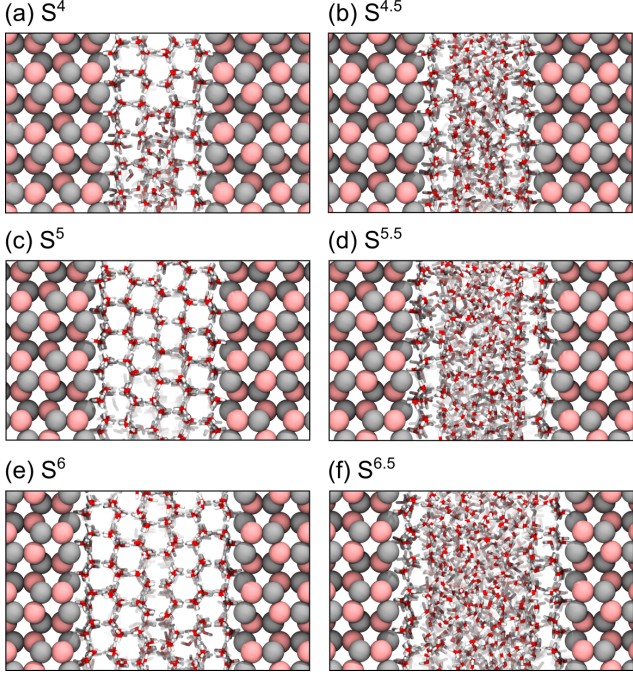

**Figure 4.** Snapshot details of the last frame of TIP4P/Ice simulations of AgI slit systems with gap widths of (a) 4 ice bilayers, (b) 4.5 ice bilayers, (c) 5 ice bilayers, (d) 5.5 ice bilayers, (e) 6 ice bilayers and (f) 6.5 ice bilayers. Ag and I are shown in silver and pink, respectively, and water molecules are shown as red and white sticks.

accelerated in the nanogrooves when the gap width is near some specific multiples of the ice lattice constant. However, both studies were constrained to certain discrete gap widths by the lattice constant of the respective surface material. Cao et al. (2019)
studied ice nucleation of mW water in slits between parallel plates of graphene and found that for all gap widths considered, freezing temperatures were above the one on the flat, hydrophobic, graphene surface, but did not observe a clear sign of ice nucleation suppression for gap widths that were non-integer multiples of the ice bilayer width. In the present work, the use of similar slit systems instead of grooves also allows us to freely tune the gap widths and to perform simulations on a much finer grid using the mW model, and to confirm these results with the atomistic TIP4P/Ice model. While we find the maximum
nucleation rates at gap widths equal to integer multiples of the ice bilayer width, we also observe enhanced nucleation rates (within one order of magnitude of the maximum enhancement) for gap widths that are within ±0.25-0.30 of the exact integer multiple value. This shows that the lattice can accommodate some distortion along the axis perpendicular to the slit surface, even while the other axes are constrained by the templating effect of the strongly hydrophilic AgI(0001) surface. However, unlike Cao et al. (2019), we find suppression of ice nucleation below the rate on the single flat AgI(0001) surface at gap widths
exactly in between multiples of the ice bilayer width.



We note that for the AgI slab models employed, ice cannot grow close to the slab edges, with about 2 nm of disordered water remaining at the edges, even at gap widths equal to integer multiples of the ice bilayer width. When simulating slabs with a smaller $z$ dimension of 10 nm, we do not observe any ice growth at all, due to the effects from the edges of the slabs.

### 3.1 Impact of temperature and ice growth dynamics

Fig. 3 shows that in the slit systems with small gap widths (*e.g.*, less than 6 ice layers), the determined nucleation rates are approximately the same for $T = 267$ K, 265 K and 263 K. The nucleation rate observed on the flat AgI(0001) surface can be used to estimate the expected temperature dependence of the nucleation rate. Following the formalism by Zobrist et al. (2007), we obtain the same nucleation rate using classical nucleation theory (CNT) by fitting the effective contact angle to $25.3041°$. This yields CNT nucleation rate estimates of $1.508 \times 10^{21}$ m$^{-2}$s$^{-1}$ and $5.1182 \times 10^{16}$ m$^{-2}$s$^{-1}$ at 265 K and 267 K,

respectively, i.e., according to CNT, nucleation rates should drop by 2 and 6 orders of magnitude at 265 K and 267 K from the 263 K value, respectively. However, we observe no temperature dependence at small gap widths. This can be explained in the CNT framework by the fact that in these systems, the ice contact angle is nearly zero, and there is no nucleation barrier. The lack of a true nucleation barrier and induction time means that the determined 'nucleation rates' are governed by the rate of ice growth in the slits. The determined induction times are approximately two orders of magnitude faster using the mW model,

which is in agreement with the faster ice growth rates that this model shows, compared to the TIP4P/Ice model (Espinosa et al., 2016a). Higher growth rates in the coarse grained mW model arise from the easier reorganization of the hydrogen bond network lacking real hydrogen atoms, compared to the all-atom TIP4P/Ice model.

## 4 Ice nucleation in AgI wedge systems

While AgI slit geometries were found to enhance ice nucleation and growth at certain gap widths, ice growth out of the slit

could not be observed. In this section, we study ice nucleation in wedge systems with different angles, described in Sec. 2.3, which serve as a model for cracks or pits on ice nucleating particles. For each system, 15 independent simulations with both mW and TIP4P/Ice models were performed at temperatures of 263 K, 265 K and 267 K. In addition, for a small subset of systems using the mW model, temperatures up to 271 K were also studied.

### 4.1 AgI wedge simulations with mW model at 263 K

Ice-like structures are observed in the wedge systems already at 273 K (after equilibration) using the mW model. Whether ice continues to grow at temperatures below the melting point depends strongly on the wedge angle. In systems which show little or no disorder between ice layers growing from each side of the wedge, ice growth is continuous and detecting critical clusters is not feasible. As a result, a nucleation rate cannot be determined for these simulations. However, the growth of ice occurred at different rates in systems with different wedge angles. Figure 5(a-f) shows the average time evolution of cubic and hexagonal

ice growth in wedge systems with different angles. Ice structures were determined using the LICH-TEST algorithm (Roudsari et al., 2021).



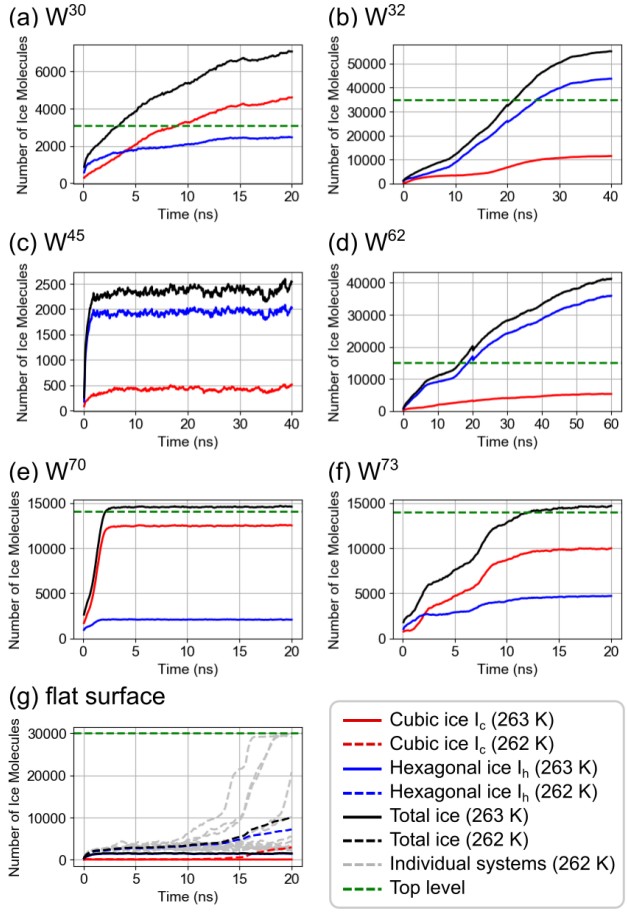

**Figure 5.** Time evolution of the average number of mW water classified as cubic or hexagonal ice and their sum in wedge systems with different angles at $T = 263$ K (a-f) and on the flat AgI(0001) surface at $T = 263$ K and 262 K (g).

Variation between the 15 individual simulations for each angle was small compared to the large differences between different wedge angles. The $W^{70}$ systems are most active for ice growth, they grow full of ice in about 3 ns. In the $W^{30}$ systems, ice grew to the top of the simulation cell within approximately 20 ns. In the $W^{73}$ systems, ice grew to fill the entire cell in less than
205 30 ns. Both systems with only one AgI(0001) surface also showed rapid ice growth, $W^{62}$ grows ice to the top of the wedge in about 15 ns, $W^{32}$ in about 20 ns. In other systems, ice cannot grow beyond formation of a few ice layers. About seven layers of ice formed at the bottom of the $W^{45}$ systems within 2 ns, with ice growing from the two sides of the wedge connecting, but we could not observe any further ice growth within 40 ns. The rest of the systems show initial ice-like structures on the two AgI(0001) surfaces, but they do not combine to form continuous ice across the wedge: $W^{60}$, $W^{110}$ and $W^{120}$ systems all show
ice slowly growing to only a maximum of three layers on each side of the wedge. Figure 6 shows simulation snapshots of ice growth in different wedge systems at time $t = 1, 2, 5$ and 10 ns.





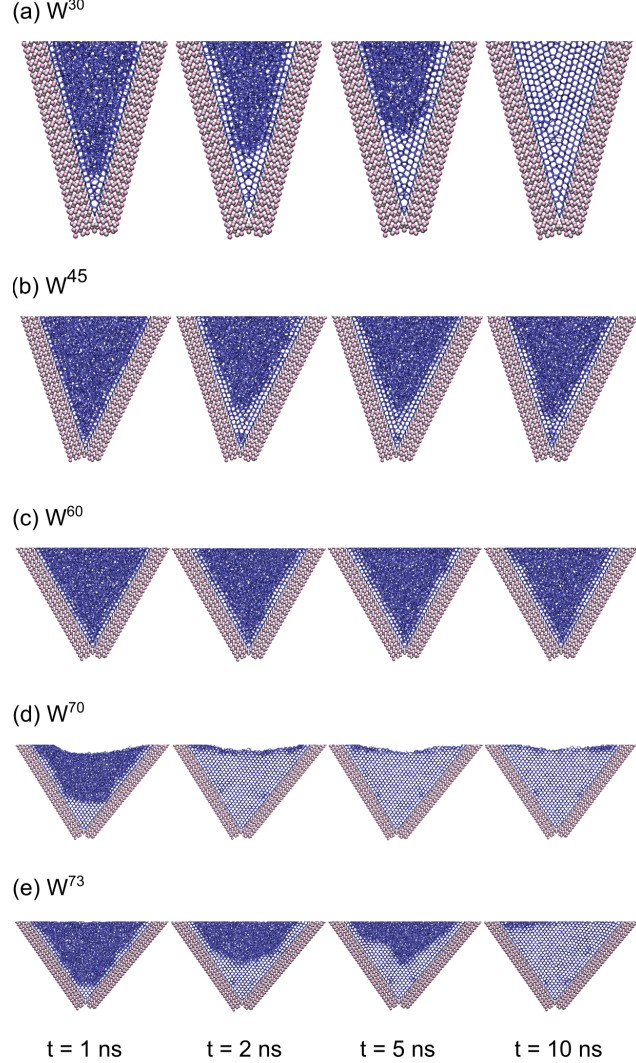

**Figure 6.** Simulation snapshot details of ice growth at times $t = 1, 2, 5, 10$ ns in wedge systems (a) $W^{30}$, (b) $W^{45}$, (c) $W^{60}$, (d) $W^{70}$, and (e) $W^{73}$, using the mW model. Ag and I are colored in silver and pink, respectively, and the hydrogen bond network between mW water molecules is indicated by blue sticks.

In general, our mW simulation results showed that the wedge systems with open angles ($W^{110}$ and $W^{120}$) have an insignificant or no effect on the formation of ice. In contrast, the wedge systems with acute angles ($W^{30}$, $W^{45}$, $W^{60}$, $W^{70}$ and $W^{73}$) enhance ice nucleation significantly. However, we observed that the level of enhancement varies in these angles and it does not always increase with decreasing the angle. The extremely rapid ice nucleation and growth at a wedge angle $\theta = 70°$ has also been observed in previous studies (Bi et al., 2017; Page and Sear, 2009). The microscopic growth mechanisms are discussed in more detail in section 4.4.





## 4.2 Nucleation of ice on flat AgI(0001) with mW model

To quantify the enhancement of ice nucleation in confined wedge geometries, we compare our results to simulations on the
220 flat AgI (0001) surface, which has been extensively studied using the TIP4P/Ice water model (Fraux and Doye, 2014; Zielke
et al., 2014, 2015; Glatz and Sarupria, 2016; Sayer and Cox, 2019; Roudsari et al., 2020). Our results with the TIP4P/Ice model
show a high nucleation rate at 263 K, consistent with previous work, but no nucleation at 265 K or 267 K at timescales easily
accessible with direct MD simulations. As there is no literature on ice nucleation simulations using the mW water model on
AgI surfaces, we studied ice nucleation on a periodic 10.1 nm × 10.3 nm perfect $\beta$-AgI(0001) surface at 260 K, 262 K, 263 K,
265 K and 267 K. At 263 K, the temperature at which most of our wedge systems nucleate ice, none of the 15 individual
simulations show ice growing beyond 3-4 ice-like layers on top of the flat surface. To rule out artefacts caused by periodic
boundary conditions or the interaction cutoff, tests with changed parameters were made, but those showed the same results.
We assume this is due to stress in the ice lattice caused by the small mismatch of lattice constants of ice $I_h$ and $\beta$-AgI(0001).
Naturally, simulations at 265 K or 267 K did not exhibit further ice growth. At 262 K, however, the surface is active in MD
timescales, with 6/15 simulations showing ice growth either to the top of the liquid film, or continuous growth towards the top,
within 20 ns, as shown in Fig. 5g.

## 4.3 AgI wedge simulations with TIP4P/Ice model at 263 K

We conducted 15 independent simulations using the TIP4P/Ice water model for wedge systems $W^{30}$, $W^{45}$, $W^{60}$, $W^{70}$, $W^{73}$,
$W^{110}$ and $W^{120}$. Unlike in the mW simulations, where ice structures grow continuously from the start of the simulation,
simulations with the TIP4P/Ice model exhibit finite induction times, and a meaningful nucleation rate can be determined from
ensemble averaging. For a given simulation system, the nucleation rate is obtained from the probability of finding the system
in the purely liquid state after time $t$, which is modelled using (Cox et al., 2015)

$$P_{\text{liq}}(t) = \exp(-Rt)^{\gamma}, \tag{1}$$

where $R$ is the nucleation rate and $\gamma$ is a compressed delay fitting parameter. The induction times are determined by monitoring
the size of the largest ice cluster in the systems using the LICH-TEST algorithm (Roudsari et al., 2021). The probability
distributions $P_{\text{liq}}(t)$ for wedge systems with different angles using the TIP4P/Ice model are illustrated in Fig. 7, and the
corresponding nucleation rates are summarized in Tab. 1. Note that the nucleation rate for these systems is given in units of
inverse time and distance, because the wedges are infinitely periodic along the $y$ direction.

As can be seen in Tab. 1, the nucleation rates obtained for $W^{30}$, $W^{60}$ and $W^{73}$ are considerably higher than for the flat
surface. Moreover, in the 15 simulations for $W^{110}$ and $W^{120}$, ice nucleation occurred only four and three times, respectively,
which is insufficient for determining a reliable nucleation rate. The simulations using TIP4P/Ice yielded similar results as with
mW for most of the systems. However, we observed differences in some of the cases. In contrast with our mW simulation
results, the $W^{60}$ and $W^{70}$ systems yielded the highest and the lowest nucleation rates, respectively, using the TIP4P/Ice model.
We discuss this observation in detail in the following section. Examples of TIP4P/ice structure in some of the wedge systems
are shown in Fig. 8.





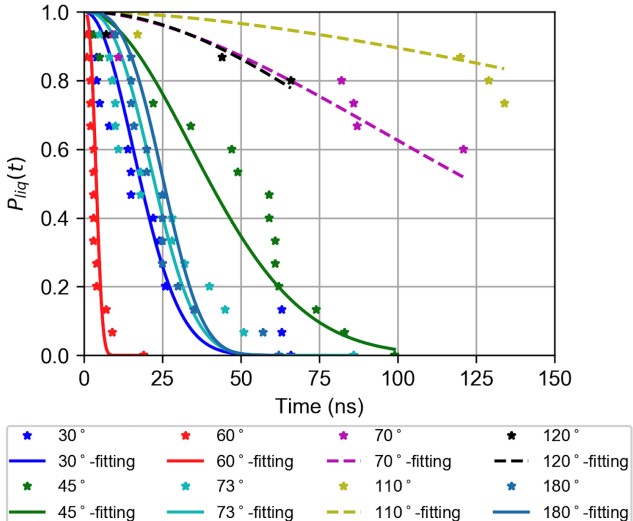

**Figure 7.** Probability for different wedge systems to remain in the liquid state as a function of time from 15 independent molecular dynamics simulations at $T = 263$ K (circles), and fits to Eq. (1) (full and dashed lines).

**Table 1.** Ice nucleation rate $R$ on different wedge systems using the TIP4P/Ice model.

| System | $R\,(\times 10^{15}\ \mathrm{m^{-1}s^{-1}})$ |
|---|---|
| $W^{30}$ | 6.16 |
| $W^{45}$ | 2.81 |
| $W^{60}$ | 30.69 |
| $W^{70}$ | 0.89 |
| $W^{73}$ | 5.24 |
| flat surface | 3.37 |

## 4.4 Microscopic ice growth mechanism

To understand how the wedge systems enhance ice nucleation, we compare the microscopic details of the ice formation in the wedge systems using both the mW and the TIP4P/Ice models. In general, the results show that confinement enhances ice nucleation. Similar to the slit systems, the confinement effect does not simply depend on the opening angle $\theta$, but more

importantly on how well an undistorted ice lattice can fit in the wedge. In addition, positions of the water molecules at the interface need to be compatible with the lattice of the AgI surfaces. Due to small differences in water bonding between the water models, this compatibility is optimal at slightly different wedge angles for the mW and the TIP4P/Ice models.

    For the TIP4P/Ice model, we observed that the $W^{30}$, $W^{60}$, and $W^{73}$ systems lead to an enhanced nucleation rate compared to the flat surface. However, tighter angles do not always yield the highest nucleation rate. In $W^{60}$ systems, which clearly





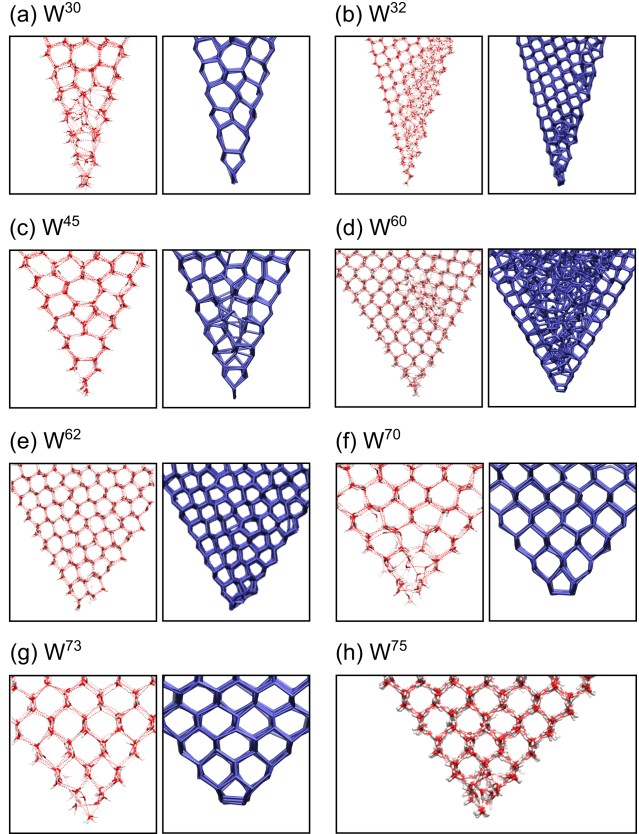

**Figure 8.** Examples of atomistic details of ice structures at the bottom of the wedge systems for the TIP4P/Ice and the mW water models for (a) $W^{30}$, (b) $W^{32}$, (c) $W^{45}$, (d) $W^{60}$, (e) $W^{62}$, (f) $W^{70}$, (g) $W^{73}$, (h) $W^{75}$. TIP4P/Ice molecules are shown as red and white sticks, and hydrogen bonds are shown as dashed red lines. For the mW model, the hydrogen bond network is shown as blue sticks.

exhibited the highest nucleation rate (see Table 1), a distorted 6-ring structure instantly forms at the bottom of the wedge which triggers the growth of ice along the two sides with a nearly perfect ice structure. However, further away from the wedge bottom, the two growing ice fronts do not coincide in the middle. As a result, defected ice structures are created a few nanometers above the bottom of the wedge, but ice continues to form around it. This can be clearly seen in the snapshots presented in Fig. 8d. In the $W^{30}$ system, with the second highest nucleation rate, the available space cannot contain a perfect ice lattice and the growth of continuous ice is here enabled by an ordered network of 5- and 7-ring structures, as shown in Fig. 8a. The existence of these geometrical defects has been shown to slow down nucleation (Donadio et al., 2005), and they have been also previously observed in graphene wedge systems (Bi et al., 2017). The $W^{73}$ system, with the third highest nucleation rate despite its large opening angle, is an example of a wedge system which accommodates a network of almost perfect ice with negligible curvature (see Fig. 8g). The nucleation rate in $W^{45}$ systems is slightly lower than on the flat surface, and the same applies to $W^{30}$, $W^{60}$, and $W^{73}$. Ice structures in $W^{45}$ always contain stacked 5- and 7-rings similar to $W^{30}$, but for this system the ring network is





also curved due to the wedge angle (see Fig. 8a and c). The $W^{70}$ system only shows nucleation in four out of 15 simulations. In those cases where ice nucleation occurred, we observe a strongly curved ice network above disordered molecules at the bottom. An example of this is shown in Fig. 8f. For the obtuse angles $W^{110}$ and $W^{120}$, we observe up to three layers of ice forming separately on each side, but they are unable to grow together and fill the wedge on the timescale of the simulation. Ice

nucleation also occurs in the wedge systems cut out of a single crystal of $\beta$-AgI, $W^{32}$ and $W^{62}$. These systems expose one (0001) surface (on the left side in Fig. 8b and e) and one higher-index surface. The $W^{62}$ system (see Fig. 8e) grows nearly perfect ice with deformed 6-rings along the higher-index surface and some curvature in the stacking of the basal ice layers. The $W^{32}$ system (see Fig. 8b) shows ice growth from the AgI(0001) side only, but the growth is slowly progressing towards the other, higher-index surface.

For the mW model, the visualization of the simulation snapshots show a very good geometrical match between the wedge systems and ice structure in $W^{70}$ and $W^{73}$ (see Fig. 8f and g), which is consistent with the growth rate results in Tab. 2. $W^{30}$ enables continuous growth of ice via a network of $5+7$ rings, similar to TIP4P/Ice (see Fig. 8a). $W^{45}$ shows some similar $5+7$ ring structures near the bottom of the wedge, but no ice growth above these structures can be observed (see Fig. 8c). The mW simulation of $W^{60}$ (see Fig. 8d) shows no ice growing inside the wedge. This can be explained by the fact that there is a

mismatch between the orientations of the ice structures starting to grow from the AgI surfaces; unlike $W^{30}$ or $W^{45}$, this angle does not accommodate any regular network of defected mW ice structures. The systems cut out of the single crystal, $W^{32}$ and $W^{62}$ (see Fig. 8b and e), exhibit similar behaviour as the simulations using TIP4P/Ice, and show the second and third highest growth rates of the mW simulations, respectively.

Looking at $W^{60}$ systems in Fig. 8d, clear differences between the results obtained using the mW and the TIP4P/Ice models

can be observed. While $W^{60}$ grows ice exceedingly well using TIP4P/Ice, despite the stacking faults in the center of the wedge, it cannot grow ice at all using mW, due to the lattice mismatch mentioned previously. Similarly, the regular network of $5+7$ defects can only be seen in the TIP4P/Ice simulation of $W^{45}$, but not when using the mW model (see Fig. 8c).

Another obvious difference between the results of mW and TIP4P/Ice simulations can be seen for the $W^{70}$ systems (see Fig. 8f): while for mW we observe a perfect ice network above the 5-ring at the bottom of the wedge, for TIP4P/Ice we observe

a strongly curved ice network above disordered molecules at the bottom in those cases where ice nucleated at all. For $W^{73}$, both mW and TIP4P/Ice can form quite well-ordered ice structures, without significant curvature. At an angle $\theta = 75°$, TIP4P/Ice can also form ice structures without curvature, but the structures at the bottom of the wedge are more disordered (see Fig. 8h).

The differences between ice structures observed using the two water models are probably mostly due to the differences in the water-AgI interaction: in the all-atom model, explicit hydrogen atoms allow more realistic interactions between water

and surface ions, and the model reproduces a more realistic behaviour of bulk water by incorporating long-range electrostatic forces (Espinosa et al., 2016b), compared to the coarse grained mW model. While in the mW model the interaction between a water molecule and Ag and I ions is described by an isotropic LJ potential only, the orientation of the atomistic TIP4P/Ice water molecule at the interface is affected by the Coulomb interactions between ions and the water molecule's partial charges, which in turn affects hydrogen bonding to neighboring water molecules.



### 4.5 Ice cubicity in wedge simulations

Besides the confinement effect, ice nucleation can also be enhanced by structural matching between wedge systems and ice structures. The Ag-terminated (0001) surface of $\beta$-AgI serves as a template for the basal (0001) plane of hexagonal ice and the (111) plane of cubic ice. For a wedge system to promote ice nucleation through structural matching, the wedge angle and the dihedral angle of the ice crystal planes must be equal or close. The basal planes of hexagonal ice grow in parallel. As a result, a $\beta$-AgI wedge system cannot promote hexagonal ice through structural matching. However, the (111) planes of cubic ice intersect at a $70°$ angle. Thus, an AgI wedge with a matching angle ($W^{70}$) can promote cubic ice, as clearly shown in our mW simulation results (see Fig. 8f). A similar effect was observed also in TIP4P/ice systems with slightly wider angles, $W^{73}$ and $W^{75}$ (see Fig. 8g and h). In both cases, ice structures at the bottom and on the sides of the wedge are mainly cubic. The ice nucleation enhancement through structural matching in wedge systems with angle $70°$ has also been observed in the work of Bi et al. (2017).

We determine the cubicity, i.e., the ratio of cubic ice to total amount of ice, for each wedge system simulation from the ice structure analysis using the LICH-TEST algorithm (Roudsari et al., 2021). In the growth of unconfined ice, stacking-disordered ice forms with random sequences of cubic and hexagonal ice, with cubicity between one-half and two-thirds(Hudait et al., 2016). In contrast, most of the wedge systems strongly favor the growth of one of the phases, and there is little variance in cubicity between samples of the same wedge type. Average ice growth rates during monotonous growth phase for each wedge type and cubicity at the end of the growth phase are reported in Tab. 2.

For each wedge system, the correlation between cubicity and growth rate was calculated as

$$\text{corr}_{CR} = \frac{\langle (C - \mu_C)(R_{\mathrm{g}} - \mu_R) \rangle}{\sigma_C \sigma_R}, \qquad (2)$$

where $R_{\mathrm{g}}$ is the growth rate during the monotonous ice growth phase for each simulation, $C$ is the cubicity of ice in each simulation at the end of the growth phase, $\mu_C$ and $\mu_R$ are the mean values and $\sigma_C$ and $\sigma_R$ the standard deviations of cubicity and growth rate for the wedge type, respectively, and the mean $\langle \dots \rangle$ is taken over the 15 simulations for each wedge type.

In the mW water model simulations with a wedge opening angle $\theta > 30°$, systems with a high growth rate show the expected positive correlation of growth rate with the majority phase increase, i.e. for mostly cubic ice containing systems, growth of cubic ice correlates positively with high growth rate, and vice versa, as shown in Tab. 2. For the $W^{30}$ system, however, the correlation is strongly negative, despite average cubicity close to one-half, and this system shows by far the largest variance in both cubicity and growth rate between simulations.

Analysis of cubicity in the TIP4P/Ice simulations yields the same majority ice type for each wedge angle as the mW results, except for $W^{32}$, but the absolute values of average cubicity can differ between mW and TIP4P/Ice simulations. Similarly to the case, For $W^{60}$ and $W^{73}$, we observe high cubicities of 0.928 and 0.990, respectively, as well as high nucleation rates, using the TIP4P/Ice model, in agreement with the above finding that ice growth is easy, when there are no changes between cubic and hexagonal ice types. In the $W^{62}$ system, both water models show low cubicity and a similar stacking disordered structure near the bottom of the wedge: one hexagonal ice layer in contact with the AgI(0001) surface, followed by some cubic ice layers before further mostly hexagonal ice growth.





**Table 2.** Ice growth rates $R_g$, cubicity $C$, and their correlation corr$_{CR}$ in different wedge systems of angle $\theta$ using the mW water model, as well as the cubicity $C_{\text{TIP4P/Ice}}$ observed using the atomistic TIP4P/Ice model. Uncertainty values show the standard deviation (instead of the usual standard error of the mean), applied in the normalization of the calculation of correlation. The average cubicity using the TIP4P/Ice model $C_{\text{TIP4P/Ice}}$ is given as comparison.

| $\theta$ (deg) | $R_g$ ($\times 10^{12}$ molecules/s) | $C$ | corr$_{CR}$ | $C_{\text{TIP4P/Ice}}$ |
|---|---|---|---|---|
| 30 | $0.440 \pm 0.304$ | $0.548 \pm 0.287$ | $-0.85$ | $0.590$ |
| 32 | $1.580 \pm 0.438$ | $0.206 \pm 0.040$ | $-0.15$ | $0.629$ |
| 62 | $1.299 \pm 0.369$ | $0.163 \pm 0.029$ | $-0.73$ | $0.327$ |
| 70 | $5.367 \pm 0.426$ | $0.853 \pm 0.004$ | $0.08$ | $0.585$ |
| 73 | $1.111 \pm 0.265$ | $0.682 \pm 0.008$ | $0.12$ | $0.990$ |

### 4.6 Nucleation and growth at lower supercooling

In our simulations at higher temperatures of $T = 265$ K and 267 K, ice nucleation was not observed on the flat AgI(0001) surfaces with either of the water models. However, we observed ice nucleation in several wedge systems. This shows that confinement has a more considerable effect on ice nucleation activity at low supercooling.

Simulations using the TIP4P/Ice water model at 265 K showed that $W^{60}$ leads to the highest nucleation activity. In particular, ice nucleation was observed in all $W^{60}$ simulations with an average nucleation time of 9.7 ns (15 independent simulations were 345 carried out for each wedge system at each temperature). The $W^{73}$ simulations resulted in nucleation in 12 systems with average nucleation time of 39 ns. The $W^{30}$ system also resulted in nucleation in 12 simulations, with average nucleation time of 77 ns. With other wedge angles, nucleation was rarely observed. Specifically, the $W^{110}$ systems nucleated in only 4 simulations with average nucleation time of 100 ns. The $W^{70}$ and $W^{45}$ systems only nucleated once, with nucleation times of 78 ns and 110 ns, respectively.

At 267 K, ice nucleation was observed only in $W^{60}$ and $W^{73}$ systems, where the $W^{60}$ system resulted in nucleation in all 15 simulations with average nucleation time of 6.4 ns, and the $W^{73}$ system nucleated only in 2 simulations with average nucleation time 131 ns.

Simulations with the mW model showed that $W^{70}$ is the most active system at 265 K, resulting in ice growth to the top of the wedge within a few nanoseconds in all 15 simulations. The $W^{30}$ system also grows ice nearly to the top of the wedge 355 within 20 ns. In $W^{62}$ systems, ice nucleates immediately starting from the bottom, forming a hemisphere up to the top corner in about 30-50 ns, with 9-12 ice bilayers counted from the (0001) surface, but never grows beyond the corner. In this system, ice is almost fully hexagonal, as the angle between the wedge sides accommodates this structure nearly perfectly. In the $W^{32}$ system, ice grows in all 15 simulations in 5.5-14 ns to 8.5-10 nm from the bottom of the wedge. In the $W^{73}$ system, ice rapidly expands from the bottom of the wedge in 12 of 15 simulations, growing 3-5 nm of ice to about half-way from the bottom of 360 the wedge in 2-16 ns.





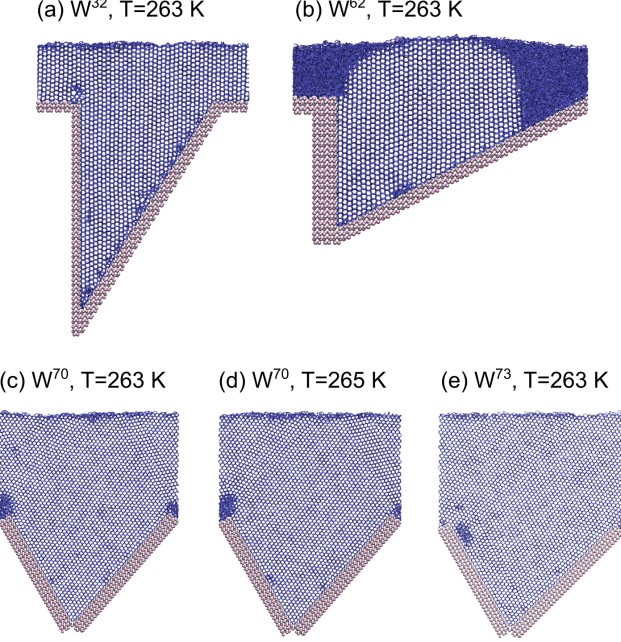

**Figure 9.** Simulations of ice outgrowing wedge systems at two temperatures using the mW model. (a) $W^{32}$ and (b) $W^{62}$ at $T = 263$ K, (c) $W^{62}$ at $T = 263$ K, (d) $W^{62}$ at $T = 265$ K, and (e) $W^{73}$ at $T = 263$ K. Ag and I ions are shown in silver and pink, respectively, and the hydrogen bond network between mW water molecules is shown by blue sticks.

At 267 K, $W^{70}$ is the only system where ice grows to the top of the wedge, whereas the other systems active at 265 K only grow 2-4 nm of ice from the bottom of the wedge. Some ice growth is observed with $W^{70}$ up to 270 K, where about 4 nm of ice grows in less than 10 ns.

### 4.7 Growth out of the wedge defects

For any surface feature to function as an ice nucleation active site, it is necessary that ice not only nucleates inside the defect, but can also grow out of the defect at a temperature above the temperature at which the flat surface or other surface features can trigger nucleation. To study this effect, additional simulations were performed with an increased amount of mW water in the simulation cell to allow ice growth out of the wedge defects. Similar simulations were not carried out using the TIP4P/Ice water model, because of the excessive computational cost of the all-atom model for the very large system sizes and simulation

370 times required. Figure 9 shows final frames of simulations of $W^{32}$, $W^{62}$, $W^{70}$ and $W^{73}$ systems, where ice is clearly growing out of the wedge defects. These wedge systems exhibit the four fastest initial growth rates when using the mW model. We note that the upper corners of the wedges are perfectly sharp, and inhibit ice growth directly at the corners. Nevertheless, ice can grow around the disordered structures at the corners, and more realistic corner geometries should therefore be an even smaller obstacle to ice outgrowing the wedge, especially considering the large surface area of ice present at this stage of growth.


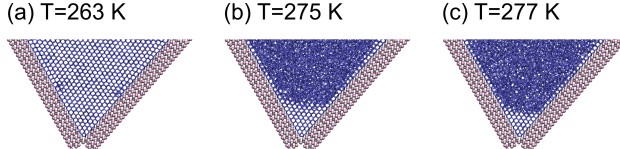

(a) T=263 K  (b) T=275 K  (c) T=277 K

**Figure 10.** Simulations of ice in the $W^{70}$ wedge system persisting above the melting point using the mW model. Last frame at (a) $T = 263$ K at time $t = 10$ ns, (b) $T = 275$ K and (c) $T = 277$ K at time $t = 40$ ns after heating to this temperature gradually from $T = 263$ K. Ag and I ions are shown in silver and pink, respectively, and the hydrogen bond network between mW water molecules is shown by blue sticks.

## 4.8 Ice persisting in wedges above the melting point

Figure 10 shows snapshots of stable ice structures at 275 K and 277 K, after heating the fully frozen $W^{70}$ system from 263 K to these temperatures, and simulating their evolution for 40 ns. While the amount of ice in the topmost layer is fluctuating, the other layers are stabilized by the structural support for the ice lattice from the side walls of the wedge. Also the $W^{73}$, $W^{62}$ and $W^{30}$ systems show some remaining ice at the bottom of the wedges at these temperatures. For the $W^{70}$ system, when cooling below 273 K, ice readily grows starting from the residual ice volume observed at higher temperature. These wedge systems could therefore initiate ice growth at even lower supercooling than when starting from purely liquid water, when temperatures fluctuate around the melting point.

## 5 Conclusions

We have studied the nucleation and growth of ice in wedge and slit geometries, exposing the Ag-terminated AgI(0001) surface using molecular dynamics simulations with the atomistic TIP4P/Ice and the coarse-grained mW models of water. We have shown that in general, confining water can significantly enhance ice nucleation even at very low supercooling, and lead to ice nucleation at higher temperatures than any known flat surfaces. However, ice nucleation enhancement, or inhibition, critically depends on the gap width of the slit structures, and the opening angle of the wedge structures.

For the slit systems, we find enhanced nucleation for gap widths accommodating integer multiples of the ice bilayer width. This enhancement effect becomes less pronounced with larger gap widths. In contrast, gap widths that differ from an integer multiple suppress nucleation, unless the gap is large enough to allow the formation of two separate ice systems on each surface. Simulations using TIP4P/Ice and mW water yield qualitatively similar results, while actual rates differ significantly. Our results agree with an earlier computational study investigating the freezing of mW water in hydrophobic slit pores (Cao et al., 2019). We found no evidence of temperature dependence in the nucleation or growth rates, suggesting that strictly speaking this is not a nucleation process.

For the wedge systems, we find enhanced or suppressed nucleation depending on the opening angle $\theta$. Whether or not a wedge structure can easily accommodate an ice structure depends on the three dimensional structural match between cubic and/or hexagonal ice structures, or regular defects such as $(5+7)$ rings, and the positions of Ag ions on the surfaces, in addition





to the wedge opening angle. This issue has also been recently investigated for several systems by Soni and Patey (Soni and
Patey, 2021). Comparison of our results in the wedge systems to the earlier study with graphite surfaces (Bi et al., 2017) found
similar favourable angles, especially for the simulations with the mW model, even though the AgI and graphene surfaces are
quite different in how strongly they pattern water into ice-like structures (although both pattern the hydration layer to match
the structure of the basal plane of ice), and even the defect structures which allow ice to form for the $W^{30}$ wedge system are
the same. The fact that the results with the TIP4P/Ice model do not agree with the results of Bi et al. (2017) as well as the mW
results is a consequence of the orientational order enforced by the surface ions on the atomistic water model, absent in mW.

While several slit and wedge systems show ice nucleation at very low supercooling, we find one critical difference: while
ice can grow out of some of the wedge systems, allowing them to function as active sites for macroscopic ice nucleation, in the
slit systems we observed no ice growth out of the confined space, and thus the slit geometry cannot help macroscopic growth
of ice, at least for the $\beta$-AgI(0001) systems considered here.

In conclusion, in addition to lowering the free energy barrier for nucleation by decreasing the ice-water interfacial area,
confinement can lead to enhancement of ice nucleation by two atomistic mechanisms: i) a suitable confining geometry provides
space where ice lattice fits (nearly) perfectly, and water molecules are effectively forced to locations matching the ice lattice.
This occurs both in the slit system with the slit gap width matching the height of an integer number of ice bilayers, and in wedge
systems, where the space accommodates a wedge-shaped piece of crystalline ice. In some cases like the $W^{30}$ system, where
the available space does not match a perfect ice lattice, the same mechanism can occur incorporating suitable regular defects
(in this case 5- and 7-rings) within the lattice. ii) Templating of water by the confining surfaces can force water molecules near
the surface into ice-like positions. For $\beta$-AgI(0001) surfaces this effect is very strong. If this templating, in combination with
a suitable geometry, causes ice-like structures from different sides of the confined space to join, i.e. there is good structural
matching, an ice nucleus of several ice layers was shown to remain stable even above the thermodynamic melting point of ice.

Of these mechanisms, i) is ultimately the dominant one: If the available space between confining surfaces cannot accom-
modate either a perfect, or *regularly* defected, ice lattice, ice growth cannot continue. This is found to be the case both in slit
geometries and in several of the wedge systems. Mechanism ii) is also very important, as the templating by AgI surfaces causes
the nearest hydration layers to form ice-like structure at very low supercooling, and ice growth can progress to fill the space
between the surfaces with ice. When we compare to an earlier study on graphene surfaces (Bi et al., 2017), we see similar
patterns of ice nucleation enhancement, but at a clearly higher temperature, facilitated by the much more favourable templating
by the AgI surfaces compared to graphene surfaces. Therefore we consider that the effects i) and ii) can be additive: the spatial
templating on top of individual surfaces combined with the possibility to fit a regular ice lattice into the confined volume cre-
ates *structural matching* and enhances nucleation. We therefore conclude that the ice nucleation enhancement by confinement
is active at a wide range of supercooling, and propose that it is relevant for different materials and surface chemistries that can
act as templates for ice faces to a varying degree.

In the context of atmospheric ice nucleating particles, our results strongly support the experimental evidence for the im-
portance of surface features such as cracks or pits functioning as active sites for ice nucleation. We have shown that suitable





geometries can enhance ice nucleation not only within the confined region, but also enable the required ice growth out of the surface features.

Finally, we find that heating some of the wedge systems above the thermodynamic melting point of ice after ice formation does not lead to full melting of ice structures. The near-perfect structural matching leads to stabilization of some ice even at these temperatures and ice is found to regrow rapidly from these structures upon re-cooling. We propose this as a very effective mechanism for ice nucleation and growth in natural wedge-like active sites, somewhat analogous to pore condensation freezing.

*Code and data availability.*   Simulation data and input files can be made available upon reasonable request. The LICH-TEST algorithm (Roud-
sari et al., 2021) used to identify ice structures is openly available on GitHub: https://github.com/opakarin/lich-test.

*Author contributions.*   GR, OHP and HV planned the research. GR and OHP carried out simulations. GR, OHP, and BR analysed the data. GR wrote the first draft, and all authors contributed to the final manuscript.

*Competing interests.*   The authors declare that they have no conflict of interest.

*Disclaimer.*   Any opinions, findings, and conclusions or recommendations expressed in this material are those of the authors and do not
necessarily reflect the views of the National Center of Meteorology, Abu Dhabi, UAE, funder of the research.

*Acknowledgements.*   This work was supported by the ERC Grant 692891-DAMOCLES, the Academy of Finland Flagship funding (grant no. 337549), the University of Helsinki, Faculty of Science ATMATH project, and the National Center of Meteorology (NCM), Abu Dhabi, UAE, under the UAE Research Program for Rain Enhancement Science. Supercomputing resources were provided by CSC-IT Center for Science, Ltd., Finland, and the Finnish Grid and Cloud Infrastructure (urn:nbn:-fi:research-infras-2016072533) supported this project with
computational and data storage resources.



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
