# Peer review of "Atomistic and coarse grained simulations reveal increased ice nucleation activity on silver iodide surfaces in slit and wedge geometries"

_Atmospheric Chemistry and Physics, 2021_

## Author Comment (AC1)

**Atomistic and coarse grained simulations reveal increased ice nucleation activity on silver iodide surfaces in slit and wedge geometries – Response to RC1**

Golnaz Roudsari, Olli Pakarinen, Bernhard Reischl, Hanna Vehkamäki

May 2022

**General comment:**

This study presents molecular dynamics simulations to investigate ice nucleation within slits and wedges of silver-terminated AgI (0001) surfaces. Ice formation depending on slit width and the opening angle of wedges was investigated. Moreover, the potential of ice to grow out of the slits and wedges was assessed. Simulations were carried out with the coarse-grained mW and the all-atom TIP4P/Ice models. It was found that slit systems promote ice nucleation when the slit width matches an integer number of ice bilayer thickness. Yet, the ice was not able to grow out of the slits. In the case of wedges, a high sensitivity to the opening angle was found with some angles inhibiting ice formation compared to the flat AgI surface and other ones enhancing it. Interestingly, the angles that enhanced or inhibited ice formation were not the same for the two water models. Moreover, ice was able to grow out of the wedges. This study exemplifies how surface geometry and templating of ice by the confining surfaces may act together to enhance ice nucleation. The differences in ice nucleation efficiency depending on the water model is well discussed and shows the potential and limitations of such simulations. Overall, it would be helpful if simulations that are described but not shown in the paper would be made accessible as supplementary information. Apart from that, this paper is well written and can be recommended for publication subject to minor revisions.

**Response**: We first would like to thank the referee for the thorough and very positive review of our manuscript and the detailed comments and questions, which will be addressed point by point below.

**Specific comments:**

**Comment 1**: method section: information or references should be given about how cubic, hexagonal, and liquid water molecules are discriminated for the two water models.

**Response**: a new subsection (2.4) has been introduced to the methods section describing the LICH-TEST algorithm employed to classify water molecules as liquid, cubic ice, or hexagonal ice. The explanations given are as follows.

Lines 140-144 (in the revised manuscript): "We use the LICH-TEST algorithm (Roudsari, 2021) for the recognition of liquid water, cubic and hexagonal ice structures, as well as different interfacial

[Figure]

Figure 1: Snapshots of the final configurations of ice nucleation simulations at $T = 263$ K using the TIP4P/Ice water model in AgI slit systems with a gap width of 5 (left) and 6 (right) ice bilayers. Ag and I are shown in silver and pink, respectively, and water molecules are shown as red and white sticks. In both systems, ice cannot grow close to the slab edges.

structures. The LICH-TEST algorithm analyses the local structure around each water molecule by identifying the number of staggered and eclipsed conformations between two neighboring water molecules, based on a template matching approach."

**Comment 2**: line 168: what is meant by "strongly hydrophilic" in terms of contact angle?

**Response**: we have rewritten that sentence to clarify our statement:

Lines 182-184: "This shows that the lattice can accommodate some distortion along the axis perpendicular to the slit surface, even though the AgI(0001) surface acts as a strong template for water molecules in the first hydration layer, in the directions parallel to the surface."

**Comment 3**: lines 171–173: It would be interesting to see the disorder of water molecules at edges in a figure as supplementary information.

**Response**: following the suggestion made by the reviewer, final snapshots of two slit systems where the effects of the slab edges can be clearly seen have been added to the supplementary information (see Fig. S1 in the SI).

In addition, final snapshots of simulations of slit systems at T= 265 K and 267 K have been added to to the supplementary information (see Fig. S2 and Fig. S3 in the SI).

The figures added are shown here in Figs. 1-3.

**Comment 4**: figures 5 and 6 should correspond better with each other. Wedges with angles that are not shown in these figures should be shown as supplementary information. The meaning of "Top level" should be explained. For some wedge angles, the total number of ice molecules exceeds

[Figure]

Figure 2: Snapshot details of the last frame of TIP4P/Ice simulations at 265 K for AgI slit systems with gap widths of (a) 4 ice bilayers, (b) 5 ice bilayers, (c) 6 ice bilayers, (d) 7 ice bilayers, (e) 8 ice bilayers, (f) 9 ice bilayers, (g) 10 ice bilayers, (h) 11 ice bilayers and (i) 12 ice bilayers. Ag and I are shown in silver and pink, respectively, and water molecules are shown as red and white sticks.

the number of ice molecules at the top level. This should be explained. Also the total number of water/ice molecules in each simulation should be stated.

**Response**: the results for $W^{60}$ have been added to Fig. 5 in the revised manuscript. In addition, we provided the simulation snapshots details of ice growth for $W^{32}$ and $W^{62}$ in Fig. 6 in the revised manuscript.

The explanation of "top level" has been added to the caption of fig. 5 in the revised manuscript:

"Top level indicates when the ice front has grown up to the top of the AgI wedge structure. In some systems, ice growth out of the wedge can be observed".

In some systems, ice growth out of the wedge. Thus, the number of ice molecules in the system become larger than the number of ice molecules inside the wedge (below top level).

The total number of water molecules in each simulation using both TIP4P/Ice and mW model are reported in Tables S1 and S2 in SI.

The updated figures are shown here in Fig. 5 and Fig. 4. The added Tables are shown here in Table 1 and Table 2.

**Comment 5**: lines 203–205: Here, it is stated that for the $W^{30}$ system ice grew to the top of the simulation cell within approximately 20 ns, but the green dashed line is already reached after less than 5 ns. This should be explained. Moreover, in the next sentence, it is stated that in the $W^{73}$ system, ice grew to fill the entire cell in less than 30 ns, but the simulation only goes to 20 ns and the green dashed line is crossed already after about 12 ns. This also requires clarification.

[Figure]

Figure 3: Snapshot details of the last frame of TIP4P/Ice simulations at 267 K for AgI slit systems with gap widths of (a) 4 ice bilayers, (b) 5 ice bilayers, (c) 6 ice bilayers, (d) 7 ice bilayers, (e) 8 ice bilayers, (f) 9 ice bilayers, (g) 10 ice bilayers, (h) 11 ice bilayers and (i) 12 ice bilayers. Ag and I are shown in silver and pink, respectively, and water molecules are shown as red and white sticks.

Table 1: Number of water molecules and simulation box dimensions used in ice nucleation simulations on wedge systems using the TIP4P/Ice water model.

| Systems | Number of water molecules | box size nm$^3$ |
|---|---|---|
| $W^{30}$ | 8883 | $9.45 \times 7.33 \times 25$ |
| $W^{45}$ | 17440 | $14.51 \times 7.33 \times 25$ |
| $W^{60}$ | 19840 | $23.95 \times 7.33 \times 25$ |
| $W^{70}$ | 8338 | $9.87 \times 7.33 \times 20$ |
| $W^{73}$ | 8342 | $10.50 \times 7.33 \times 20$ |
| $W^{110}$ | 8068 | $11.87 \times 7.33 \times 20$ |
| $W^{120}$ | 7957 | $12.35 \times 7.33 \times 20$ |

**Response**: as explained in our response to the previous comment, the green dashed lines in Fig. 5 of the manuscript indicate the top level of the wedge systems while lines referred in the comment discuss the time when whole system freezes. We rewrote the paragraph to discuss the times of ice reaching the top of the AgI wedges, consistently with Fig. 5 in the revised manuscript:

Lines 218-222: "Ice grows in these systems to the top of the wedge in about 2-3 ns. In the $W^{30}$ systems, ice grew to the top of the wedge within approximately 3 ns. In the $W^{73}$ systems, ice grew to the top of the wedge in about 12 ns. Both systems with only one AgI(0001) surface also showed rapid ice growth, $W^{62}$ grows ice to the top of the wedge in about 15 ns, $W^{32}$ in about 20 ns."

**Comment 6**: lines 206–207: "About seven layers of ice formed at the bottom of the $W^{45}$ systems within 2 ns." Does this statement refer to Fig. 6b? If yes, a reference to this figure could be given

[Figure]

Figure 4: Time evolution of the average number of mW water classified as cubic or hexagonal ice and their sum in wedge systems with different angles at $T = 263$ K (a-g) and on the flat AgI(0001) surface at $T = 263$ K and 262 K (h). Top level indicates when the ice front has grown up to the top of the AgI wedge structure. In some systems, ice growth out of the wedge can be observed.

here.

**Response**: the references to the figure in the comment (now Fig.6c) and for $W^{60}$ system (Fig.6d) have been added to the revised manuscript. Thank you.

**Comment 7**: line 209: The systems $W^{110}$ and $W^{120}$ should be shown as part of SI.

**Response**: as suggested by the reviewer, the snapshots of systems $W^{110}$ and $W^{120}$ for mW simulations have added to the SI (see Fig. S5 in the SI). See Fig. 6 here.

**Comment 8**: lines 212–213: Do you mean this statement in absolute terms or relative to the flat AgI surface?

**Response**: we agree that there are ambiguities. The statements are relative to the flat AgI surface. The point in the comment has been addressed in the revised manuscript:

Lines 227-232: "In general, our mW simulation results showed that the wedge systems with open

(a) W$^{30}$        (d) W$^{60}$

(b) W$^{32}$        (e) W$^{62}$

(c) W$^{45}$        (f) W$^{70}$

(g) W$^{73}$

t = 1 ns    t = 2 ns    t = 5 ns    t = 10 ns        t = 1 ns    t = 2 ns    t = 5 ns    t = 10 ns

Figure 5: Simulation snapshot details of ice growth at times $t = 1, 2, 5, 10$ ns in wedge systems (a) $W^{30}$, (b) $W^{32}$, (c) $W^{45}$, (d) $W^{60}$, (e) $W^{62}$, (f) $W^{70}$, and (f) $W^{73}$, using the mW model. Ag and I are colored in silver and pink, respectively, and the hydrogen bond network between mW water molecules is indicated by blue sticks.

Table 2: Number of water molecules and simulation box dimensions used in ice nucleation simulations on wedge systems using the mW water model

| Systems | Number of water molecules | box size nm$^3$ |
|---|---|---|
| $W^{30}$ | 17766 | $9.54 \times 7.33 \times 18$ |
| $W^{32}$ | 63661 | $17.99 \times 10.08 \times 29$ |
| $W^{45}$ | 17440 | $14.54 \times 7.33 \times 15.4$ |
| $W^{60}$ | 30775 | $24 \times 7.33 \times 17$ |
| $W^{62}$ | 63280 | $22.48 \times 10.08 \times 16.1$ |
| $W^{70}$ | 18056 | $19.1 \times 7.33 \times 15$ |
| $W^{70}_{outgrowing}$ | 72236 | $19.1 \times 7.33 \times 33.5$ |
| $W^{73}$ | 19205 | $19.5 \times 7.33 \times 15$ |
| $W^{73}_{outgrowing}$ | 77784 | $22.97 \times 7.33 \times 27$ |
| $W^{110}$ | 33009 | $31.4 \times 7.33 \times 15$ |
| $W^{120}$ | 22824 | $30.1 \times 7.33 \times 13$ |

angles ($W^{110}$ and $W^{120}$) have an insignificant or no effect on the formation of ice. In contrast, several wedge systems with acute angles ($W^{30}$, $W^{32}$, $W^{62}$, $W^{70}$ and $W^{73}$) enhance ice nucleation significantly compared to the flat AgI (0001) surface. However, we observed that the level of enhancement varies in these angles and it does not necessarily increase with decreasing the angle, and

[Figure]

W$^{110}$

W$^{120}$

Figure 6: Simulation snapshot details of the last frame ($t = 150$ ns) for wedge systems $W^{110}$ (top) and $W^{120}$ (bottom) at $T = 263$ K, using the mW model. Ag and I are colored in silver and pink, respectively, and the hydrogen bond network between mW water molecules is indicated by blue sticks.

$W^{45}$ and $W^{60}$ showed no particular ice growth activity."

**Comment 9**: line 213: why is $W^{45}$ mentioned in the bracket? There seems to be hardly any ice formation for this opening angle.

**Response**: the reviewer is correct, we apologize for the confusion. We removed $W^{45}$ and $W^{60}$ from the list of effective wedge angles and instead discuss them separately. $W^{32}$ and $W^{62}$ were added to the list of effective wedge angles (see the revision in the response to the previous comment).

**Comment 10**: line 226–227: the statement "none of the 15 individual simulations show ice growing beyond 3–4 ice-like layers on top of the flat surface" does not become evident based on Fig. 5g. Again, a supplementary figure would be helpful.

**Response**: in the revised manuscript, the results for flat surface at T= 263 K and 262 K have referenced to Fig.5h (solid black line), and Fig.5h (dashed black line), where the number of ice molecules are shown. In addition, snapshots for the simulations at T= 263 K and 262 K have been provided in SI (see Fig. S4 in the SI). See Fig. 7 here.

**Comment 11**: line 239: what is a "compressed delay fitting parameter"? An explanation and/or a reference should be given.

**Response**: we revised the explanation to Line 255 in the revised manuscript: "... and gamma is a delay parameter that determines the induction time". In addition, the reference has been provided (Cox et al., 2015) in the manuscript.

[Figure]

Figure 7: Simulation snapshot details of maximum ice growth observed in ice nucleation simulations on the flat AgI(0001) surface using the mW model (a) at $T = 263$ K in 60 ns, and (b) at 262 K in 20 ns. Ag and I are colored in silver and pink, respectively, and the hydrogen bond network between mW water molecules is indicated by blue sticks.

**Comment 12**: line 244: Table 1 shows an enhancement in nucleation rate of less than a factor of two for $W^{30}$ and $W^{73}$ compared with the flat surface. This seems to me only a minor increase in nucleation rate. What is your criterion for a "considerable" increase?

**Response**: we agree that the description was inaccurate. This has been addressed in the revised manuscript:

Lines 260 and 261: "As can be seen in Table 1, the nucleation rate for $W^{60}$ is significantly higher than for the flat surface, and also $W^{30}$ and $W^{73}$ show somewhat higher nucleation rates".

**Comment 13**: line 273–274: again, these simulations could be shown as part of SI.

**Response**: as suggested by the reviewer, the snapshots of systems $W^{110}$ and $W^{120}$ for TIP4P/Ice simulations have been added to the SI (see Fig. S6 and Fig. S7 in the SI).

See Fig. 8 and Fig. 9 here.

**Comment 14**: lines 301–304: What does it mean for the predictive power of MD simulations when they are so sensitive to the specific water model? An additional comment would be helpful.

**Response**: in the revised manuscript, a comment on the correspondence and complimentary characteristics of mW and TIP4IP/Ice has been added at the end of Sec. 4.4. The comment is as follows:

Lines 321-323: "While the overall agreement between the main ice nucleation simulation results with the coarse-grained mW and the atomistic TIP4P/Ice models indicates that our findings are robust, the differences observed also highlight the importance of checking coarse-grained simulation results with an atomistic model whenever possible."

**Comment 15**: lines 333–334: Should "Similarly to the case," be deleted?

**Response**: the text has been corrected. Thank you.

**Comment 16**: The content of Table 2 should be explained better.

[Figure]

Figure 8: Simulation snapshot details of the last frames of 15 independent simulations of wedge systems $W^{110}$ at $T = 263$ K using the TIP4P/Ice model. Ag and I are shown in silver and pink, respectively, and water molecules are shown as red and white sticks.

[Figure]

Figure 9: Simulation snapshot details of the last frames of 15 independent simulations of wedge systems $W^{120}$ at $T = 263$ K using the TIP4P/Ice model. Ag and I are shown in silver and pink, respectively, and water molecules are shown as red and white sticks.

**Response**: Table 2 has been changed in the revised manuscript: we now refer to the wedge systems in the common way ($W^{\theta}$), instead of specifying the value of $\theta$ in the table. We have removed the

redundant second reference to $C_{TIP4P/Ice}$. Otherwise, we believe that all quantities presented in the table are correctly mentioned in the table caption and defined or explained sufficiently in Sec. 4.5.

In the revised manuscript, the caption of Table 2 now reads: "Ice growth rates $R_g$, cubicity $C$, and their correlation $corr_{CR}$ in different wedge systems using the mW water model, as well as the cubicity $C_{\text{TIP4P/Ice}}$ observed using the atomistic TIP4P/Ice model. Uncertainty values for the mW results show the standard deviation (instead of the usual standard error of the mean), applied in the normalization of the calculation of correlation".

**Comment 17**: Line 343: Which angles have been investigated at 265 K? A summarizing table with all results from both models (how many simulations produced ice and in what average time?) would be helpful.

**Response**: We have created Table S3 of different wedge systems at $T = 263$ K, 265 K and 267 K, with induction times for each individual simulation using TIP4P/Ice water model in the SI (see Table 3 here). In addition, we have added Table S4 with a summary of observed nucleation events using mW water model at $T = 265$ K and 267 K in the SI (see Table 4 here). The simulation results using mW model at 263 K are explained in detail in Section 4.1 in the manuscript. The average nucleation times are reported in Section 4.6 of the revised manuscript.

Table 3: Induction times for ice nucleation (ns) for 15 independent simulations of different AgI wedge systems at temperatures 263 K, 265 K and 267 K using the TIP4P/Ice water model. Dashes (-) indicate *no nucleation* event within 150 ns.

| Systems | T (K) | Simulations | | | | | | | | | | | | | | |
|---|---|---|---|---|---|---|---|---|---|---|---|---|---|---|---|---|
| | | 1 | 2 | 3 | 4 | 5 | 6 | 7 | 8 | 9 | 10 | 11 | 12 | 13 | 14 | 15 |
| $W^{30}$ | 263 | 15 | 63 | 22 | 24 | 5 | 8 | 14 | 25 | 26 | 21 | 4 | 2 | 63 | 66 | 4 |
| | 265 | 95 | 50 | 122 | - | - | 116 | 132 | 12 | 54 | 119 | 18 | 85 | 93 | 30 | - |
| | 267 | - | - | - | - | - | - | - | - | - | - | - | - | - | - | - |
| $W^{45}$ | 263 | 74 | 5 | 34 | 59 | 99 | 61 | 49 | 3 | 83 | 61 | 9 | 62 | 59 | 22 | 47 |
| | 265 | - | - | - | - | - | - | - | - | - | - | 110 | - | - | - | - |
| | 267 | - | - | - | - | - | - | - | - | - | - | - | - | - | - | - |
| $W^{60}$ | 263 | 2 | 3 | 7 | 2 | 2 | 4 | 4 | 3 | 3 | 3 | 9 | 19 | 3 | 2 | 5 |
| | 265 | 12 | 10 | 6 | 23 | 3 | 26 | 4 | 9 | 8 | 6 | 8 | 12 | 9 | 7 | 3 |
| | 267 | 11 | 5 | 2 | 5 | 9 | 4 | 3 | 3 | 4 | 2 | 5 | 3 | 6 | 27 | 7 |
| $W^{70}$ | 263 | - | 121 | - | - | - | 82 | 86 | - | 11 | - | - | 9 | - | - | 87 |
| | 265 | - | - | 78 | - | - | - | - | - | - | - | - | - | - | - | - |
| | 267 | - | - | - | - | - | - | - | - | - | - | - | - | - | - | - |
| $W^{73}$ | 263 | 10 | 28 | 10 | 32 | 8 | 5 | 18 | 51 | 45 | 9 | 86 | 28 | 11 | 18 | 40 |
| | 265 | 20 | 43 | 7 | - | 17 | 3 | 59 | - | - | 74 | 61 | 37 | 15 | 13 | 119 |
| | 267 | - | - | 129 | - | - | - | 133 | - | - | - | - | - | - | - | - |
| $W^{110}$ | 263 | - | - | - | - | 17 | 120 | - | - | - | - | - | 134 | - | - | 129 |
| | 265 | - | - | - | - | - | - | - | - | - | - | - | - | - | - | - |
| | 267 | - | - | - | - | - | - | - | - | - | - | - | - | - | - | - |
| $W^{120}$ | 263 | - | - | 66 | - | - | 7 | - | - | - | 44 | - | - | - | - | - |
| | 265 | - | - | - | - | - | - | - | - | - | - | - | - | - | - | - |
| | 267 | - | - | - | - | - | - | - | - | - | - | - | - | - | - | - |

**Comment 18**: Line 362–363: Movies as part of SI could illustrate this statement.

Table 4: Fraction of simulations exhibiting an ice nucleation event for different AgI wedge systems at temperatures 265 K and 267 K using the mW water model.

| Systems | T (K) | Number of nucleation events |
|---|---|---|
| $W^{30}$ | 265 | 15/15 |
| | 267 | 0/15 |
| $W^{32}$ | 265 | 15/15 |
| | 267 | 0/15 |
| $W^{62}$ | 265 | 15/15 |
| | 267 | 0/15 |
| $W^{70}$ | 265 | 15/15 |
| | 267 | 15/15 |
| $W^{73}$ | 265 | 12/15 |
| | 267 | 0/15 |

[Figure]

Figure 10: Simulations of ice outgrowing wedge systems at $T = 263$ K, and the effect of temperature on the ice growth in the $W^{70}$ system using the mW model. (a) $W^{32}$ and (b) $W^{62}$ at $T = 263$ K, (c) $W^{73}$ at $T = 263$ K, (d) $W^{70}$ at $T = 263$ K, (e) $W^{70}$ at $T = 265$ K, (f) $W^{70}$ at $T = 267$ K, and (g) $W^{70}$ at $T = 270$ K. Ag and I ions are shown in silver and pink, respectively, and the hydrogen bond network between mW water molecules is shown by blue sticks.

**Response**: in the revised manuscript, we have added snapshots for $W^{70}$ at T= 267 K and 270 K to Fig. 09 (panels f and g), to fully illustrate the discussion in the text (see Fig. 10 here). Hopefully, this is satisfactory for the reviewer. Please see the modifications to the text in the response to the next comment.

**Comment 19**: Line 364–365: This sentence is unclear. The formulation should be improved.

**Response**: we revised the explanation to lines 381-383: "At 267 K, $W^{70}$ is the only system where ice grows to the top of the wedge (see Fig. 9f), whereas the other systems active at 265 K only grow 2-4 nm of ice from the bottom of the wedge. The $W^{70}$ system exhibits ice formation even at

temperatures up to 270 K, where we observed the growth of about 4 nm of ice at the bottom of the wedge within 10 ns (see Fig. 9g)"

**Technical comments**:

Line 66: "Sects." Instead of "Sect."

Line 68: the point is missing at the end of the sentence.

Line 73: a comma is missing after (Abascal et al., 2005).

Figure 3: purple and blue colors are very similar and difficult to discriminate. Consider to replace e.g. purple by red.

Line 211: "times" instead of "time".

Lines 244 and 321: "Table" should not be abbreviated by "Tab.".

Figure caption of Fig. 7: "circles" appear to be stars.

**Response**: We would like to thank the reviewer again for the careful review of our work. All the technical comments are addressed in the revised manuscript. Regarding Figure 3, we have changed the colors to blue and green, to achieve better contrast and readability (see Fig. 11 here).

[Figure]

Figure 11: Nucleation rates in AgI slit simulations and inverse of time at which full ice growth in the AgI slit is observed, at (a) 263 K, (b) 265 K, and (c) 267 K, with mW model (continuous line) and TIP4P/Ice model (bar plot). Blue crosses indicate TIP4P/Ice simulations in which nucleation was not observed. The nucleation rate at 263 K on the flat AgI(0001) surface using the TIP4P/Ice model is indicated by the dashed black line in panel (a).

---

## Author Comment (AC2)

**Atomistic and coarse grained simulations reveal increased ice nucleation activity on silver iodide surfaces in slit and wedge geometries – Response to RC2**

Golnaz Roudsari, Olli Pakarinen, Bernhard Reischl, Hanna Vehkamäki

May 2022

**General comment:**

This in an interesting paper that illustrate some of the complexities of heterogeneous ice nucleation on AgI by showing how the rate of nucleation in slits and wedges depends on the slit width and wedge angle. In a similar way to how AgI can enhance nucleation by providing a local matching template for ice growth, slits and wedges that provide a good match (e.g. by having a slit width that matches an integer number of ice bilayers) to the structure of the resulting ice crystal also enhance nucleation. I am happy to recommend publication once consideration has been given to the technical issues mentioned below.

**Response**: We thank the referee for the very positive review of our work.

**Specific comments:**

**Comment 1**: polar surfaces such as the AgI (0001) surface are well-known to be unstable in the absence of a polarity compensation mechanism. By keeping the ions in the AgI crystal fixed, this instability is avoided (albeit perhaps artificially), but it is important to consider how the electrostatic boundary conditions might affect the observed properties. Some discussion of these questions would be appreciated. The Sayer and Cox paper already referenced and J. Chem. Phys. 153, 164709 (2020) are interesting in this regard.

**Response**: We thank the referee for the good comment. In the absence of trustworthy ab initio MD data or experimental atomic resolution images of the interface, the stability of the polar (0001) surface of AgI remains somewhat of a mystery. The available empirical force fields [e.g. Rains et al., PRB 44, 17, 9228 (1991)] cannot be used to model bulk-terminated AgI(0001) in periodic boundary conditions without constraining the surface and compensating the dipole field. The approach employed by Sayer and Cox, while both elegant and physically sound, does not lend itself to the present study, where we need to investigate a large number of systems with different symmetries. However, in the slit systems the dipoles cancel out, and even in the asymmetric wedge systems, the dipole field components perpendicular to the surface are at least reduced. We have added a short paragraph in the methods section to discuss this problem:

Lines 80-86 (in the revised manuscript): "An artificial constraint is usually necessary as the polar

surface is unstable using conventional force fields fitted to reproduce properties of bulk systems, and rigid surfaces have been employed in the majority of computational studies of silver iodide (Fraux and Doye, 2014; Zielke et al., 2014; Zielke et al., 2015; Glatz et al., 2016; Roudsari et al., 2020). Stable interfaces with unconstrained surface ions in flat polar surfaces could only be achieved by introducing counter ions in the solution and imposing electrostatic boundary conditions on the simulation box (Sayer et al. 2019, 2020)."

**Comment 2**: by keeping the ions in the AgI crystal fixed rather than allowing them to exhibit thermal vibrations around their lattice positions provides a more perfect template for ice nucleation than would be expected if vibrations were allowed. Some discussion of how this potentially affects the results would be appreciated.

**Response**: We agree with the referee's comment and have added the following sentence to the methods section. However, as stated in the response to Comment 1, we believe that there is no easy solution to avoid this problem, and while it affects the results quantitatively, we are quite confident that the qualitative differences between the systems that we have reported would not be affected by this.

Lines 85-86: "We note that suppressing the thermal motion of surface ions enhances the ordering of water at the interface and thereby affects ice nucleation rates (Fraux and Doye, 2014)."

---

## Author Response (AR2)

**Atomistic and coarse grained simulations reveal increased ice nucleation activity on silver iodide surfaces in slit and wedge geometries – Response to editor**

Golnaz Roudsari, Olli Pakarinen, Bernhard Reischl, Hanna Vehkamäki

May 2022

**comment:**

The author's statement "Simulation data and input files can be made available upon reasonable request" are not acceptable in view of ACPs Data Policy (https://www.atmospheric-chemistry-and-physics.net/policies/data-policy.html). Please indicate clearly, how readers can get access to the models, and make the simulation output used in the analysis available.

**Response**: First, we would like to thank the editor for handling the review of our work. As recommended in the comment, in the revised manuscript, we have provided a link to a dataset containing simulation input and output files, as well as the analyzed data used in the manuscript. Below is the link to the dataset.

https://doi.org/10.23729/d841cfd5-eef9-4ae4-a820-50becf91ec97.